# Deep learning detection and classification of fungal and non-fungal calcifications on paranasal sinus CT imaging

Zepa Yang[1,2], Insung Choi[2,3], Hoo Yun[2,4], Siwoo Kim[2,5], Hye Na Jung[4,6], Sangil Suh[4,6], Bo Kyu Kim[4,7], Byungjun Kim[4,7], Sung-Hye You[4,7], Inseon Ryoo[4,6]*

1 Department of Computer Engineering, Soonchunhyang University, Asan, Republic of Korea, 2 Biomedical research center, Korea University Guro Hospital, Seoul, Republic of Korea, 3 Seers Technology Co. Ltd., Gyeonggi-do, Republic of Korea, 4 Korea University College of Medicine, Seoul, Republic of Korea, 5 Department of Electrical and Computer Engineering, College of Engineering, Seoul National University, Seoul, Republic of Korea, 6 Department of Radiology, Korea University Guro Hospital, Seoul, Republic of Korea, 7 Department of Radiology, Korea University Anam Hospital, Seoul, Republic of Korea

* isryoo@gmail.com

## Abstract

This study aimed to develop and evaluate a deep learning algorithm for detecting and classifying intrasinus calcifications on paranasal sinus (PNS) computed tomography (CT) for the diagnosis of fungal sinusitis and differentiation of fungal and non-fungal sinusitis. A dataset of 277 PNS CT cases from Korea University Guro Hospital, supplemented by temporal and geographic external test sets, was utilized. A 3D U-Net model was employed to segment maxillary sinus regions. YOLO v5 identified calcifications, followed by classification into three patterns: normal sinus or chronic sinusitis without calcifications, dense peripheral dystrophic calcification, and central punctate fungal calcification. A separate convolutional neural network (CNN) refined the classification to ensure accurate categorization of calcification patterns. The 3D U-Net model achieved a Dice Similarity Coefficient of 0.9674, indicating accurate segmentation. YOLO v5 demonstrated precision of 79.50% and recall of 92.14% in detecting calcifications. The CNN classification model attained F1 scores of 94.73%, 90.60%, and 94.01%, and overall accuracies of 97.48%, 86.87%, and 94.01% for internal, temporal, and geographic test sets, respectively. This study demonstrated the capability of deep learning algorithms to accurately detect and classify fungal sinusitis-related calcifications on PNS CT scans. The developed framework achieved high accuracy in segmentation of sinus area and detection/classification of intrasinus calcifications. The framework also demonstrated its potential for broader application to radiographic imaging.

**Data availability statement:** Data cannot be shared publicly due to the confidentiality of individuals' medical records. Data are available from the Institutional Review Board of each hospital (contact via kughirb@kumc.or.kr and eirbadmin@kumc.or.kr) for researchers who meet the criteria for access to confidential data.

**Funding:** This work was supported by the National Research Foundation of Korea (Grant No. 2022R1F1A1062925 to IR), Korea Health Technology R&D Project through the Korea Health Industry Development Institute (KHIDI), funded by the Ministry of Health and Welfare, Republic of Korea (Grant No. HR20C0021 to ZY), and Soonchunhyang University Research Fund (Grant No. 2025-0044 to ZY). The funders had no role in the study design, data collection and analysis, decision to publish, or preparation of the manuscript.

**Competing interests:** The authors have declared that no competing interests exist.

## Introduction

During the past decade, artificial intelligence and deep learning technologies have been applied to various industrial fields including medicine [1–4]. As a huge amount of medical imaging data are acquired and stored every day, radiology is a promising field for deep learning technologies, and many studies have shown the variety of possibilities in this field [5–12]. Furthermore, the number of medical images continues to increase explosively due to advancements in medical imaging techniques, which has tremendously increased workloads for radiologists [13–15]. Application of deep learning technologies to radiology can be very beneficial to radiologists and can serve as a novel academic field in radiology.

It is crucial to differentiate fungal sinusitis from non-fungal sinusitis to determine the appropriate treatment strategy for chronic sinusitis. In particular, early detection of fungal sinusitis is important for the prevention of complications in immunocompromised individuals, since fungal sinusitis can progress to fatal invasive fungal infections in that population [16]. Therefore, patients scheduled to receive immunosuppressive treatment usually undergo screening paranasal sinus (PNS) computed tomography (CT) prior to the treatment to detect and treat existing fungal sinusitis.

Intrasinus calcification is a characteristic feature of fungal ball, usually aspergillosis. Approximately 69–77% of patients with aspergillosis have been reported to have intrasinus calcifications on PNS CT [17–19]. Intrasinus calcifications can also occur in non-fungal inflammatory diseases of the PNS, such as mucocele or bacterial sinusitis. However, intrasinus calcification is uncommon in non-fungal inflammatory sinonasal disease; less than 3% of cases exhibit intrasinus calcifications [17,18,20]. In addition, the shape and location of intrasinus calcifications of fungal sinusitis are different from those of non-fungal sinusitis [18].

There have been several studies applying deep learning algorithms to PNS CT images. Most of the studies evaluated the presence and severity of chronic sinusitis or sinus opacifications on PNS CT using deep learning algorithms [21–24]. Recently, a study assessed the performance of the algorithm on PNS CT for distinguishing among chronic sinusitis, fungal sinusitis, and healthy controls [25]. However, internal calcifications and their patterns were not evaluated and have largely been overlooked.

In this study, we developed a deep learning algorithm for detecting calcifications in the maxillary sinuses on PNS CT and classifying these features to diagnose fungal sinusitis, especially fungal ball. We also evaluated the accuracy of the algorithm for detecting and classifying calcifications.

## Materials and methods

### Data acquisition

A dataset comprising PNS CT images from 277 cases (554 images of sinuses) was collected from Korea University Guro Hospital (KUGH). The dataset included patients diagnosed with fungal calcifications in the maxillary sinus from January 2015 to February 2022. The diagnosis of fungal calcifications was confirmed by pathological

examination of specimens obtained from endoscopic sinus surgery. Pathologists diagnosed fungal ball with specific findings such as fungal hyphae. As PNS CT scans had bilateral maxillary sinuses, the contralateral side without calcification was also included in this study. We also collected data from 71 patients (142 sinuses) at KUGH from March 2022 to December 2023 as a temporal external test set and data from 105 patients (210 sinuses) at Korea University Anam Hospital (KUAH) from January 2021 to December 2023 as a geographic external test set. Although the images in the external test set were collected from a different branch of the same university-affiliated hospital system, the practicing physicians and radiologists, as well as the CT machines and imaging protocols, were different. Moreover, since the two hospitals are located in opposite parts of the city (northeast vs. southwest), the patient populations they serve differ significantly. We obtained approval from the Institutional Review Boards (IRBs) of KUGH and KUAH (IRB No.: 2022GR0432 and 2023AN373) and the need for informed consent was waived. The data comprised Digital Imaging and Communications in Medicine (DICOM) images, each with a resolution of 512 x 512 pixels. These images were collected under a strict protocol approved by the IRB, ensuring that all patient-related data were handled with confidentiality and integrity. The de-identified data were then used exclusively for the purpose of this study. The overall characteristics of the patient dataset are presented in Table 1.

## Development environment

The machine learning models in this study were developed and trained using the PyTorch deep learning framework (version 1.10.2). All computational experiments were conducted on a high-performance workstation equipped with an NVIDIA RTX A6000 GPU.

## 3D maxillary sinus area segmentation

The 3D maxillary sinus area segmentation model focused on processing PNS CT images to accurately delineate the maxillary sinus area. The model used 554 DICOM images from 277 patients. These images were subjected to a series of preprocessing steps to enhance the robustness and accuracy of the model. The initial preprocessing step involved

Table 1. The overall characteristics of the patient dataset. Distribution of class labels of each dataset was displayed. Left/right sinuses are counted separately.

| Characteristics | Training | Internal | Geographic | Temporal |
|---|---|---|---|---|
| Case count | 222 | 55 | 105 | 71 |
| Age (in years) | 61.7 | 63.15 | 65.47 | 67.38 |
| Female | 151 | 34 | 63 | 45 |
| Male | 71 | 21 | 42 | 26 |
| Tube voltage (kv) | | | | |
| 120 | 26 | 13 | 86 | 11 |
| 100 | 195 | 42 | 21 | 57 |
| 80 | 1 | 0 | 0 | 3 |
| Slice thickness (mm) | | | | |
| 1 | 0 | 0 | 78 | 1 |
| 2 | 0 | 0 | 28 | 0 |
| 3 | 222 | 55 | 1 | 70 |
| Labeled class | | | | |
| Class 1 | 200 (45.0%) | 50 (45.5%) | 95 (45.2%) | 67 (47.2%) |
| Class 2 | 64 (14.4%) | 9 (8.2%) | 17 (8.1%) | 9 (6.3%) |
| Class 3 | 180 (40.5%) | 51 (46.4%) | 98 (46.7%) | 66 (46.5%) |

cropping the maxillary sinus area from the images. This was achieved by bisecting the image along the central vertical line and calculating the y-axis range, ensuring the inclusion of the maxillary sinus area in all images, resulting in images of 256 x 256 pixels. To ensure consistency across the dataset, images of the left and right maxillary sinus areas were processed to maintain a standard orientation. This involved horizontally flipping the images of the left maxillary sinus, a method also serving as data augmentation. Additionally, to account for variation in patient positioning, these images were rotated within a 10–15-degree range.

The core of the segmentation model was the 3D U-Net architecture, which consisted of an encoder-decoder structure, allowing for precise localization while retaining important contextual information. The overall graphical diagram of proposed model is shown in Fig 1.

The encoder path of the U-Net began with a series of convolutional layers, each followed by Rectified Linear Unit (ReLU) activation functions and batch normalization to stabilize the learning process. The convolutional layers progressively reduced the spatial resolution of the feature maps while increasing the number of feature channels, allowing the model to capture increasingly abstract features. Each down-sampling step was implemented with 3D max-pooling operations, which reduced the dimensions of the feature maps by half, effectively capturing spatial hierarchies at different scales.

In the decoder path, the feature maps were up-sampled using transposed convolutions, reversing the resolution reduction performed in the encoder. These up-sampled feature maps were then concatenated with the corresponding feature maps from the encoder path through skip connections, ensuring that the fine-grained detail lost during down-sampling were preserved. Each up-sampling step was followed by 3D convolution layers to refine the segmented regions. The final layer of the network used a softmax activation function to generate a probability map, representing the likelihood of each voxel belonging to the maxillary sinus area.

During training, the Dice loss function was employed to optimize the network. This function measures the similarity between predicted and actual segmentation areas, ensuring that the model's predictions align closely with the true anatomical structures.

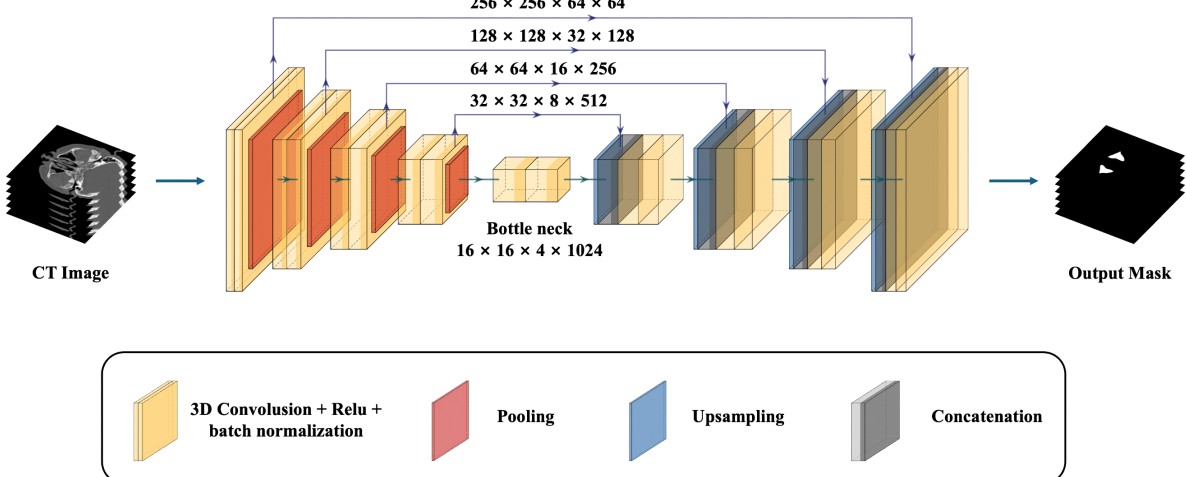

**Fig 1. Three-dimensional U-Net (N = 4 down-sampling levels) for maxillary sinus segmentation.** The encoder comprises four levels; each level applies two 3 × 3 × 3 convolutions with ReLU activation and batch normalization, followed by 3D max-pooling. Feature channels by encoder level are 32, 64, 128, and 256. The decoder mirrors the encoder with transposed convolutions for up-sampling and skip connections to the corresponding encoder features, followed by two 3 × 3 × 3 convolutions (ReLU and batch normalization). A final 1 × 1 × 1 convolution with softmax produces voxel-wise probability maps of the maxillary sinus. Training optimized a Dice-based loss with standard spatial and intensity augmentation.

## Detection and classification model for calcifications in fungal sinusitis

To facilitate the detection and classification of calcification patterns in fungal sinusitis, the input DICOM images were processed by applying a maxillary sinus mask, isolating the region of interest (ROI). The images retained the original Hounsfield unit values in the maxillary sinus region. Calcification patterns were labeled as ROIs by a head and neck radiologist, and this labeled data underwent a coordinate transformation process for use in model training.

The detection model employed YOLO(You Only Look Once) v5 to identify calcifications within the segmented maxillary sinus regions. Both the model design and the preprocessing pipeline were specifically adapted to accommodate 10-bit images, including modifications to image normalization parameters, dataset mean, and standard deviation, ensuring accurate handling of the full range of CT Hounsfield values. The training dataset consisted of 554 images of the sinuses, split into 70% for training, 10% for testing, and 20% for internal validation.

Pre-classification and labeling of the classes were based on interpretation by head and neck radiologists. Under the supervision and thorough examination of these radiologists, two radiology technicians performed labeling. Labeling focused on regions identified as fungus, with approximately 8,000 bounding boxes labeled across 554 sinuses.

During the labeling process, all YOLO-detected bounding boxes and their adjacent sinus areas were reviewed by two board-certified radiologists. Chronic or non-fungal sinusitis cases without visible calcification were categorized as Class 1 (no calcification) under radiologists' supervision. The YOLO detector was trained with a relatively low confidence threshold to minimize false negatives, ensuring that subtle or ambiguous sinus regions were proposed for subsequent classification.

In our study, the classification model recognized several distinct classes based on calcification patterns: 1) Normal sinus or chronic sinusitis, representing clear sinuses or sinus opacifications without abnormal calcifications, 2) Dense peripheral calcification (dystrophic), denoting denser calcification areas often associated with non-fungal sinusitis, and 3) Central punctate calcification patterns indicative of fungal sinusitis. Each of the calcification patterns classified in our dataset is presented in Fig 2.

During training, the Complete Intersection over Union (CIoU) loss function was used for region detection, considering the predicted bounding box's center position, size, and aspect ratio. Binary Cross-Entropy (BCE) loss function was utilized to classify the areas. These adaptations and training methods were vital in developing a model capable of accurately detecting calcification patterns in fungal sinusitis.

Additionally, a separate classification process was assigned for the bounding boxes produced by YOLO's detection results. The classification model was expected to perform multi-class classification based on calcification patterns, refining its predictions following the issues identified during the above detection stage by minimizing false positives related to artifacts.

A simple Convolutional Neural Network (CNN) with five layers was designed to classify calcification patterns in the bounding box results of the YOLO v5 model, which was assumed to reflect sinusitis with calcifications. The first convolutional layer applied 32 filters 3 x 3 in size with a stride of 1 and padding to preserve the spatial dimensions. A ReLU activation function was used to introduce non-linearity. A 2 x 2 max-pooling operation with a stride of 2 was used to reduce the spatial dimensions of the feature maps. The second convolutional layer employed 64 filters 3 x 3 in size, with similar stride and padding settings, followed by another ReLU activation function. This layer included another 2 x 2 max-pooling operation with a stride of 2. The third convolutional layer utilized 128 filters 3 x 3 in size, with a stride of 1 and padding, followed by a ReLU activation function. The output from the convolutional layers was flattened and fed into a fully connected layer with 512 neurons, followed by a ReLU activation function. The final output layer was a fully connected layer with a softmax activation function designed to output the probability distribution over the classification labels. The training dataset was augmented using random rotations, shifts, and flips to improve the model's robustness and generalization capabilities.

## Model performance evaluation

The segmentation performance in the maxillary sinus region was initially evaluated using the Dice Similarity Coefficient (DSC), a machine learning metric. This was complemented by visual assessment by head and neck radiologists. For the

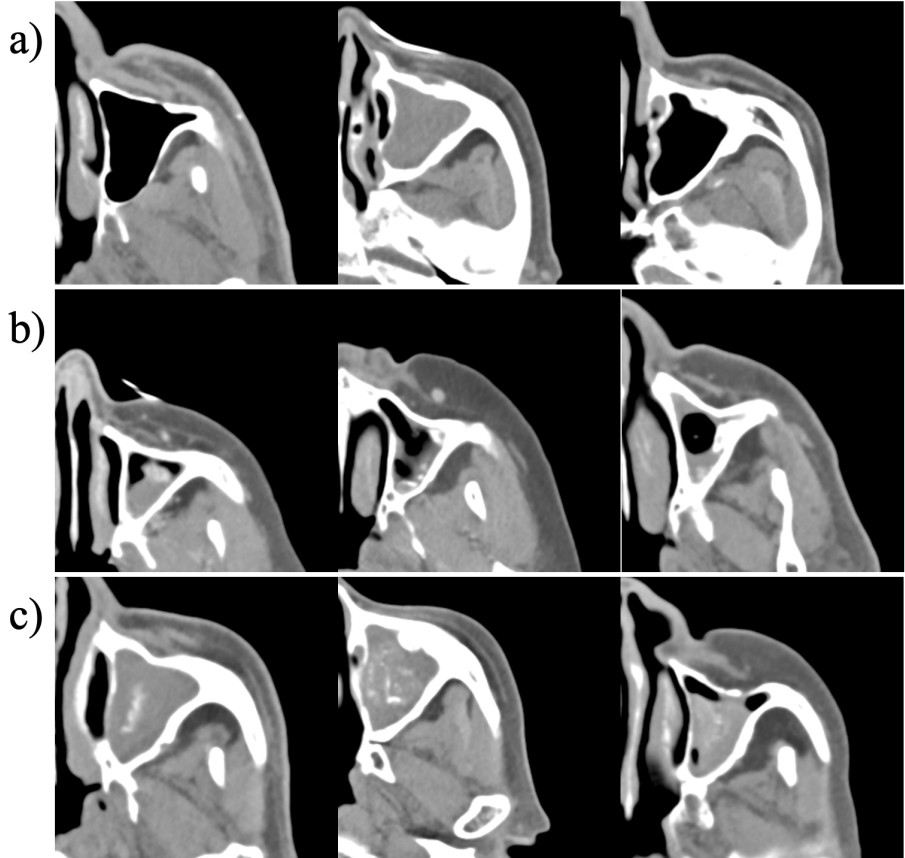

**Fig 2. A series of axial slices from PNS CT scans demonstrating maxillary sinus anatomy and calcification patterns.** a) normal sinus or chronic sinusitis, representing clear sinuses or sinus opacifications without abnormal calcifications, b) dense peripheral calcification (dystrophic), denoting denser calcification areas often associated with non-fungal sinusitis, and c) central punctate calcification pattern seen in fungal sinusitis.

detection and classification of fungal sinusitis calcifications, we employed a confusion matrix. Each patient's data were assessed and validated at the patient case-wise level, with separate evaluations for the left and right maxillary sinus regions. The gold standard was defined not solely by the conventional training criteria of the deep learning model, but by clinical calcification patterns confirmed through pathologic results.

Our classification model, trained on cases with a slice thickness of 3 mm, was also adapted for cases with a 1-mm slice thickness by dividing the slices into three sets and integrating the results. A key element of classification was identifying the presence of fungal calcification patterns in the central area of the maxillary sinus and the density of the object. The classification process was refined by evaluating the intensity brightness of the target object relative to the background, ensuring that it could be differentiated but was not significantly brighter than non-fungal dense calcification patterns. Therefore, if any slice in a patient's CT scan showed a calcification pattern in the central area and the intensity brightness met the specified criteria, the case was classified into Class 3. This classification approach was applied consistently to both 3-mm and 1-mm slice-thickness cases. Accuracy was defined as the sum of the correctly matched diagnosis and prediction results, specifically the sum of the diagonal values in the confusion matrix. The error rate was defined as the sum of all other values in the matrix, representing the total number of instances where the diagnosis and prediction results did not match. All classification metrics were reported as macro-averaged values, unless otherwise specified. Each class (normal/chronic sinusitis, peripheral calcification, and fungal punctate calcification) was equally considered in these evaluations.

For external validation with both the temporal external test set and geographic external test set, the confusion matrix was calculated, allowing for comprehensive evaluation of the model's performance across different datasets.

To evaluate the discriminative performance of the calcification classification model, receiver operating characteristic (ROC) curves were generated, and the area under the ROC curve (AUROC) was measured for each class.

We report two complementary accuracy measures. Overall accuracy (micro) is the proportion of correctly classified sinuses out of all sinuses across all classes. Balanced accuracy (macro recall) is the simple average of the recalls computed separately for Class 1, Class 2, and Class 3; it mitigates the influence of class imbalance by giving each class equal weight. We also report per-class precision, recall, and F1, as well as AUROC.

## Results

### Maxillary sinus segmentation performance

The 3D U-Net model effectively segmented the maxillary sinus regions. The model achieved a Dice loss of 0.0326 and a DSC of 0.9674, showing a high level of agreement between the predicted segmentation and the ground truth masks. Expert visual assessment of the actual segmentation results confirmed that the model produced highly accurate segmentation. Segmentation of the space between the maxillary sinus and the nasal cavity was performed, and the model also achieved relatively high accuracy in delineating the bone contours. Even in areas with complex topology and ambiguous boundaries, the model showed excellent performance. The results of the maxillary sinus area segmentation are presented in Fig 3.

### Detection and classification model for calcifications in fungal ball

The YOLO v5-based detection model effectively identified regions with suspected calcifications within the segmented maxillary sinus areas. The model achieved a CIoU loss of 0.0350 and a BCE loss of 0.0023, reflecting strong performance in both detection and localization tasks. The mAP50 (mean Average Precision) value was recorded at 0.6801, with mAP50-95 at 0.3557. The sample result of the detected bounding box in the target area is presented in Fig 4.

During training, the box loss was 0.3386, classification loss was 0.8624, and distribution focal loss (DFL) was 0.2818. Evaluation metrics indicated a precision of 0.79495 and a recall of 0.92143. While the high recall suggests that the model successfully detected most of the true positives, the relatively lower precision points to a higher occurrence of false

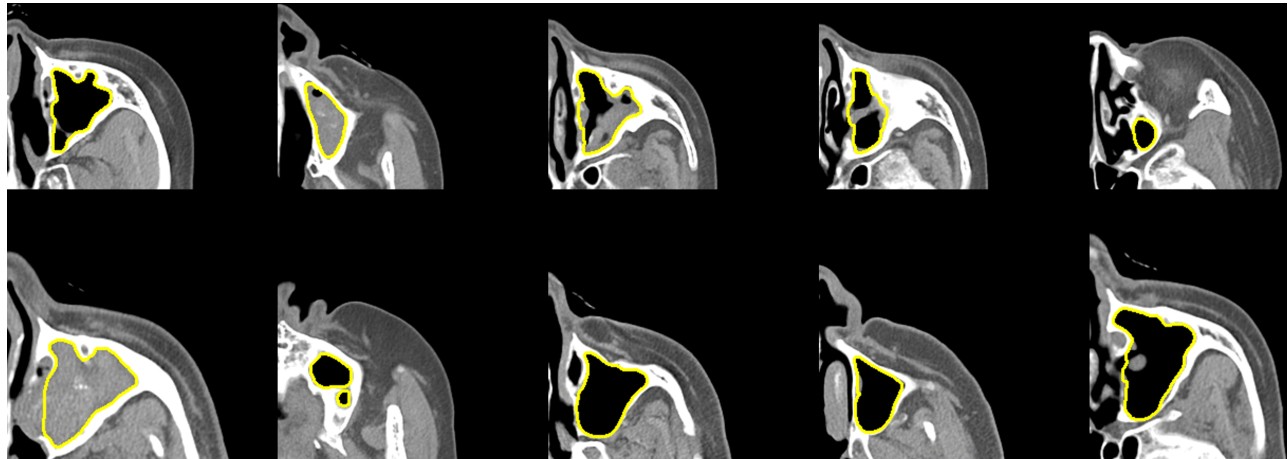

**Fig 3. Predicted results of the maxillary sinus segmentation model.** The model accurately delineated the area of interest, successfully segmenting challenging regions with ambiguous boundaries due to topological changes.

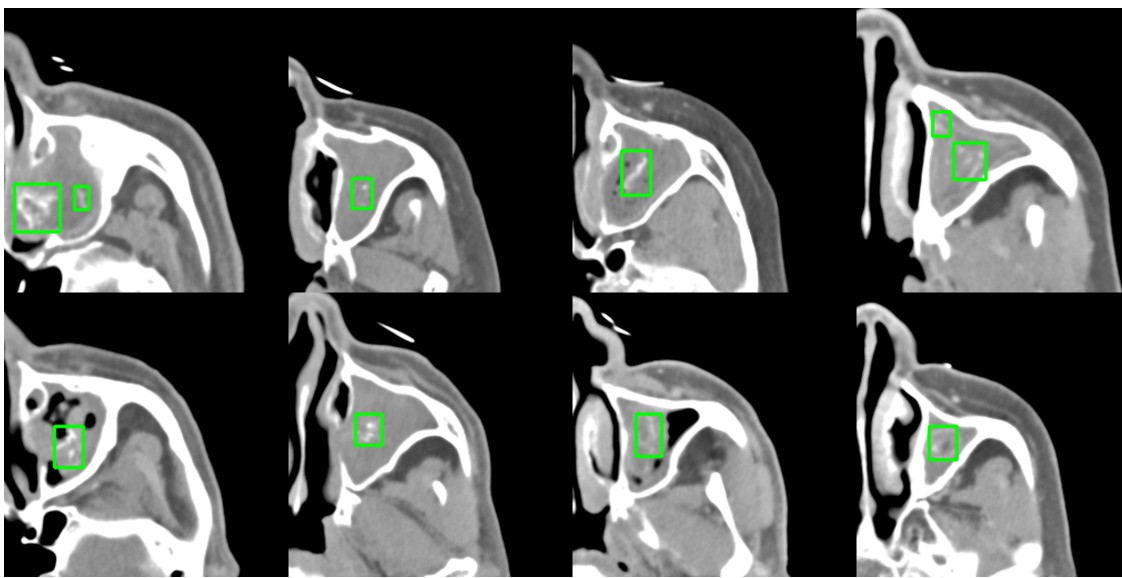

**Fig 4. Predicted results of the fungal calcifications detection model.** Regions suspected of calcifications were identified and measured using bounding boxes. While the model accurately localized the areas of interest, there was a slight tendency for higher false positive cases, such as implantation artifacts and small bone structures.

positives. The mAP50 was 0.6927, and mAP50-95 was 0.3909. In the validation phase, the box loss was 0.5653, classification loss was 0.9312, and DFL was 0.2818.

The classification model performed multi-class classification based on calcification patterns. During training, the classification model achieved an accuracy of 0.9325 and a loss of 0.1342, while in validation, it recorded an accuracy of 0.9193 and a loss of 0.1557, demonstrating high performance on unseen data. The results of overall training result are presented in Table 2.

**Table 2. Performance summary of the segmentation, detection, and classification models resulted during training process of each model. The table shows the metrics for segmentation (Dice loss, DSC), detection (CIoU loss, BCE loss), and classification (training/validation accuracy and loss), providing an overview of the models' performance.**

| Models | Score |
|---|---|
| Maxillary sinus segmentation model | |
| Dice loss | 0.0326 |
| DSC score | 0.9674 |
| Fungal sinusitis calcification pattern Detection model(YOLO v5) | |
| CIoU loss | 0.0350 |
| BCE loss | 0.0023 |
| Pattern classification model(CNN) | |
| Training accuracy/ loss | 0.9325/ 0.1342 |
| Validation accuracy/ loss | 0.9193/ 0.1557 |

DSC score = Dice Similarity Coefficient Score.

BCE loss = Binary Cross Entropy Loss.

CIoU loss = Complete Intersection over Union Loss.

## External validation and generalization

In internal validation, the algorithm achieved a macro-averaged precision of 0.9270, recall of 0.9748, and F1 score of 0.9473 across all classes. The overall accuracy was 0.9748, as shown in Table 3. These values reflect the model's ability to maintain a high balance between detecting true positives and minimizing false negatives within the training dataset. The relatively high recall of 0.9748 indicates that the algorithm performed well in identifying most positive cases, while precision at 0.9629 points to its effectiveness in minimizing false positives. The F1 score of 0.9607 confirms that the model balanced both precision and recall efficiently in the internal validation phase. The overall performance metrics of each validation set are presented in Table 3.

For external validation, the algorithm demonstrated a balanced accuracy of 95.07% for the temporal external test set and 96.19% for the geographic external test set. Confusion matrices were computed at the study level for both test sets, and F1 scores were derived using precision and recall. In the geographic external test set, the macro-averaged F1 score was 0.9401, and in the temporal external test set, it was 0.9060, reflecting robust classification performance across classes. The precision for the temporal test set was slightly lower compared to the geographic test, but recall remained high in both test sets, showing the model's consistency in detecting positive cases across different datasets.

In evaluation of the discriminative performance, the AUROC for Class 1 (0.54) was relatively lower compared to the other classes, and Class 2 and Class 3 achieved AUROC values of 0.98 and 0.93, respectively, indicating strong performance. The ROC plot is presented in Fig 5, and the confusion matrix results for each validation dataset are shown in Fig 6.

## Discussion

Most of the intrasinus calcifications in fungal sinusitis or fungal ball are centrally located in the maxillary sinus, in contrast to the calcifications in non-fungal sinusitis, which are peripherally located near the walls of the maxillary sinuses. Fungal ball usually shows fine punctate calcifications, while non-fungal sinusitis shows smooth-margined, round, or eggshell calcifications [17,18]. These different patterns of intrasinus calcifications may result from the different pathogenesis. The calcifications in fungal sinusitis are formed from metabolic deposits of calcium within the necrotic area of the mycelial

**Table 3. Performance metrics of the calcification pattern classification model across internal validation set, temporal external test set, and geographic external test set. Overall accuracy and balanced accuracy (macro recall) are reported to provide a comprehensive evaluation of the model's classification performance across fungal and non-fungal sinusitis cases.**

| Feature | Internal | Temporal | Geographic |
|---|---|---|---|
| Precision(Class 1) | 0.9630 | 0.9306 | 0.9787 |
| Precision(Class 2) | 0.8182 | 1 | 0.8824 |
| Precision(Class 3) | 1 | 0.9688 | 0.9596 |
| Recall(Class 1) | 1 | 1 | 0.9684 |
| Recall(Class 2) | 1 | 0.6667 | 0.8824 |
| Recall(Class 3) | 0.9245 | 0.9394 | 0.9694 |
| F1Score(Class 1) | 0.9811 | 0.9641 | 0.9735 |
| F1Score(Class 2) | 0.9 | 0.8 | 0.8824 |
| F1Score(Class 3) | 0.9608 | 0.9539 | 0.9645 |
| Accuracy(Class 1) | 1 | 1 | 0.9684 |
| Accuracy(Class 2) | 1 | 0.6667 | 0.8824 |
| Accuracy(Class 3) | 0.9245 | 0.9394 | 0.9694 |
| Overall accuracy | 0.9748 | 0.8687 | 0.9401 |
| Balanced accuracy | 0.9649 | 0.9507 | 0.9619 |

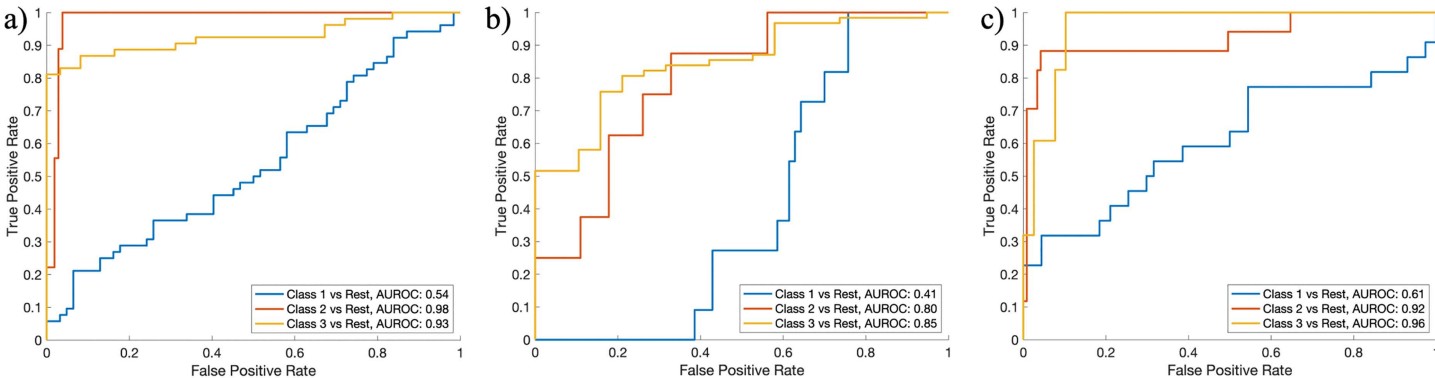

**Fig 5. Receiver Operating Characteristic (ROC) curves for multi-class classification across three experiments.** a) internal validation, b) temporal external test set, and c) geographic external test set. High discrimination performance was observed for Class 2 and Class 3, while the discrimination for Class 1 was relatively low.

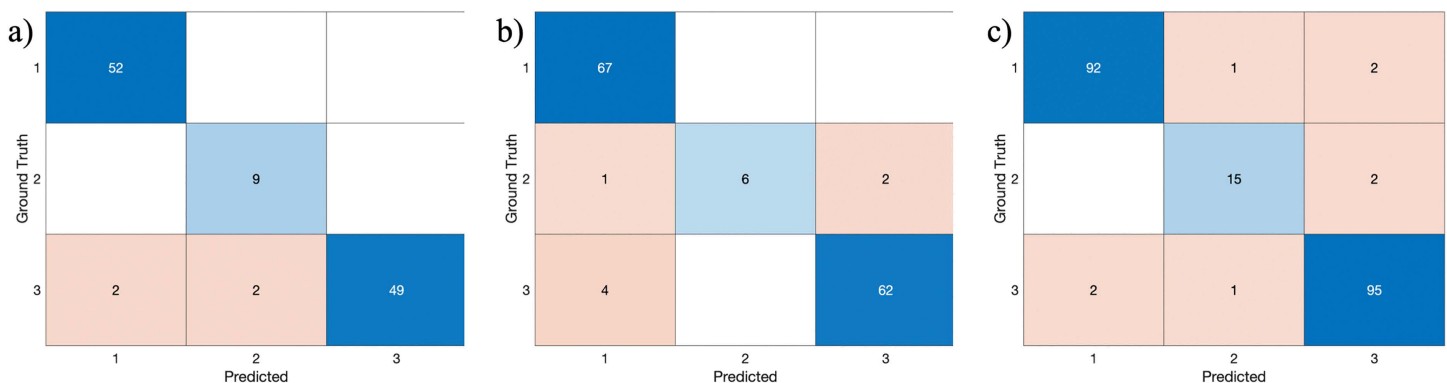

**Fig 6. Confusion matrices for a) internal validation, b) temporal external test set, and c) geographic external test set.**

mass [18,26–28]. These calcifications are located centrally, as the mycelial mass is usually located in the center of the maxillary sinus [26,27]. On the other hand, intrasinus calcifications of non-fungal sinusitis are dystrophic calcifications or ossifications caused by chronic inflammatory processes. This kind of calcification seems to occur near the thickened mucosal layer of the sinus repeatedly affected by chronic inflammation [29,30].

Because intrasinus calcifications are common in fungal ball and their shape and location are different from those of non-fungal sinusitis, detecting calcifications and classifying their patterns on PNS CT is important in the early diagnosis and treatment of fungal sinusitis.

The 3D U-Net demonstrated superior performance over the 2D U-Net for maxillary sinus segmentation. Unlike the 2D U-Net, which processes CT slices independently and may overlook spatial relationships, the 3D U-Net segments the sinus in three dimensions, capturing its structure more accurately. By leveraging information across consecutive slices, it improves boundary delineation and segmentation accuracy. These findings confirm the 3D U-Net's suitability for precise anatomical segmentation in medical imaging.

Based on the mAP and box loss values, it can be concluded that the detection performance of the YOLO v5 model was reasonably high for identifying calcified regions. However, upon visually comparing the detection result tiles with the ground truth labels, it was observed that the detection bounding boxes tended to be larger than the actual labeled regions.

We adopted a two-stage detection–classification scheme in which YOLO v5 serves as a high-recall proposal generator and a separate CNN performs fine-grained class refinement (Classes 1–3). This allowed us to run YOLO v5 with a low detection threshold to maximize sensitivity for small calcifications, while delegating false-positive reduction and subtle discrimination—especially between non-fungal and fungal calcifications—to the CNN. In preliminary checks, the single-stage YOLO head was less discriminative for these nuanced patterns, which is consistent with YOLO's emphasis on speed and objectness rather than fine-grained class separation. This design aligns with prior cascaded deep-learning frameworks for lesion detection and classification in medical imaging [31–34].

In terms of classification, the model's classification loss and precision indicated that the classification capability of the current YOLO model was not as robust as we expected. Specifically, the initial false positive rate of the YOLO model reached approximately 21.1%. The majority of these misclassifications were related to Class 2 and Class 3 images, which the model struggled to distinguish correctly. A key contributing factor to this issue was the relatively small amount of Class 2 data compared to other classes, leading to imbalanced training. Additionally, the ambiguity between Class 1 and the conventional background class further exacerbated the model's misclassification. To address these issues, a separate CNN model was trained using augmentation techniques to improve classification accuracy. This approach successfully reduced the false positive rate, as reflected in the higher precision and recall values in the results, indicating a significant improvement in model performance. The augmentation process allowed the model to better distinguish between similar classes and improve its overall classification accuracy.

In this study, the YOLO based algorithm exhibited fair detection accuracy for intrasinus calcifications, with showing BCE loss of 0.0023 and mAP50 of 0.6801 during training, and for the internal dataset, the algorithm achieved a macro-averaged precision of 92.70%, recall of 97.48%, and F1-score of 94.73%, based on performance across all three classes. The overall accuracy for the internal dataset was 97.48%, while the balanced accuracy was 96.49%. The results for temporal and geographic external test sets also showed an accuracy of around 90%. The results of our study demonstrated slightly higher accuracy compared to the findings from Kim et al., which demonstrated 88% accuracy in internal validation [25]. While Kim et al. primarily focused on distinguishing different types of sinusitis, such as fungal sinusitis, chronic sinusitis, and healthy controls, without specifically detecting calcifications, our research emphasized the detection of intrasinus calcifications.

In Kim et al.'s study [25], they could not reveal the cause of misclassification. In some cases, incorrect regions were analyzed, such as outside the sinuses, while in other instances correct regions were analyzed but still misclassified. However, as we focused on calcifications in the sinuses, we could figure out the processes the algorithm performed. First, the algorithm detected calcifications and then classified them based on their patterns. This method might also increase the accuracy of diagnosing fungal sinusitis. Even though the calcification detection rate was 91%, the classification accuracy was 97.5%, as fungal sinusitis typically led to multiple calcifications in the sinuses. Furthermore, when we evaluated the false positive cases with class 2, we found that the peripheral portion of larger calcifications or tiny calcific foci surrounding dense dystrophic calcifications might be misinterpreted as Class 3 (S1 Fig).

Detecting calcifications and classifying them based on patterns is very important in the interpretation of medical images of not only fungal sinusitis but also many other diseases as well because many diseases have unique calcifications on medical images. Some cancers including breast and thyroid cancers have unique microcalcifications and vascular diseases such as atherosclerosis and coronary artery diseases have typical vascular wall calcification patterns [35–38]. Our deep learning algorithms can be applied to those diseases as well. Furthermore, this technique could be adapted for use in radiographic imaging to identify and classify regions with specific attenuation characteristics other than calcifications. For example, by modifying the algorithm to detect areas with low attenuation, it could be applied to a wider range of conditions that involve fat, air, or other substances with low attenuation, thereby broadening its clinical utility.

Class 1 (no abnormal calcification) showed a lower AUROC (0.54) than the calcification classes. This does not contradict the confusion-matrix accuracy but reflects two design and data characteristics. First, we operated the proposal stage

at a low detection threshold to reduce missed calcifications, which increases the number of non-calcification candidates forwarded to the classifier and can raise the false-positive rate for the negative class at some thresholds, thereby depressing ROC discrimination. Second, negative patches are heterogeneous, including mucosal thickening, small bony structures, and dental or metallic artifacts, so the "no-calcification" boundary is intrinsically broader than for calcification-positive classes. In practice, class-specific operating points and probability calibration are likely to stabilize Class-1 behavior.

As summarized in Table 1, Class 2 (non-fungal calcification) is less prevalent than Classes 1 and 3 across cohorts (Training, Internal, Temporal, Geographic). Such imbalance can inflate overall or micro-averaged metrics while obscuring class-specific errors. The relative scarcity and greater phenotypic heterogeneity of non-fungal calcifications likely contribute to lower recall and PPV for Class 2 in our results. Because labels were assigned at the sinus level with left and right counted separately, a single case can contribute different classes across sides; this design increases statistical power but complicates direct comparison with case-level summaries. For prospective deployment, cohort-specific calibration and operating-point selection may be warranted to accommodate prevalence shifts between the Internal, Temporal, and Geographic cohorts.

When analyzing the multi-class confusion matrix in terms of precision and recall, it was observed that the predictive performance for internal validation data, which constituted a relatively large portion of the training dataset, was relatively strong. The model achieved high precision and recall across all classes, indicating robust performance on data similar to what it was trained on. Although the performance on geographic or temporal external test sets was slightly lower than that on the internal validation data set, it still demonstrated high accuracy.

However, there is a concern that the recall values were unusually high, especially on the internal validation set, which may indicate a potential overfitting issue. The model's tendency to classify a large number of cases as positive could indicate that it has learned patterns specific to the training data, potentially reducing its ability to generalize. This is particularly troubling given that the robustness of segmentation was not thoroughly established, yet the recall figures were still significantly elevated. The high recall might mean the model is too aggressive in identifying positive cases, which could lead to overfitting. The concern about overfitting might be acceptable to some extent since the process identifies additional regions within a segmented area rather than measuring volumetric data directly. When viewed as a pre-processing step to narrow down the search area, high recall could be justified. This approach frames segmentation as part of a broader strategy to refine target areas for further analysis. External validation using temporal and geographic test sets showed consistent performance, providing evidence against overfitting concerns. These results suggest the model captures meaningful patterns beyond the training data. Future work could address overfitting further by applying cross-validation with diverse datasets or using stricter regularization techniques during training.

There are several limitations in this study. First, 554 images represent a relatively small dataset for training a deep learning algorithm. To enhance the robustness and generalization of the model, multiple data augmentation techniques were employed, such as random rotation, scaling, and slight shifts of the original images, ensuring that the anatomical structures of the maxillary sinus remained intact. Although data augmentation techniques were employed to address data limitations, they may not fully replace an adequately large and balanced dataset. As more data is better for training an algorithm, further study with a large number of cases is needed. Second, we performed segmentation of the maxillary sinuses to facilitate calcification detection and other sinus areas were not investigated. To overcome this limitation, further research is needed, possibly involving the development or application of different models tailored to detect fungal sinusitis in additional areas. Expanding the scope of this model would provide a more comprehensive approach to diagnosing and managing fungal sinusitis across all affected sinus regions. Third, we included only non-invasive fungal sinusitis, specifically fungal ball, which typically exhibits characteristic calcification patterns on radiographic images. Allergic fungal sinusitis and invasive fungal sinusitis (both acute and chronic forms) were not included in this study. Allergic fungal sinusitis usually presents with hyperintense sinus contents, which largely overlap with findings in other inflammatory sinus diseases that contain high levels of proteinous or mucinous material. Invasive fungal sinusitis, on the other hand, is

characterized by bony destruction and direct invasion into adjacent structures. These features were not assessed in the present study. Future research incorporating these findings will be necessary to achieve a more comprehensive evaluation of all types of fungal sinus diseases. Last, the number of Class 2 cases was relatively small, especially for the temporal external test set. That lowered the accuracy of the algorithm in the external tests.

## Conclusion

This study demonstrated the effectiveness of a deep learning-based algorithm for detecting and classifying intrasinus calcifications on PNS CT to diagnose fungal sinusitis, specifically fungal ball. The developed framework achieved high accuracy in segmentation, detection, and classification of intrasinus calcifications, with consistent performance across internal and external test datasets. By focusing on calcification patterns, the proposed algorithm enhanced diagnostic precision for fungal sinusitis, highlighting its potential for broader application in medical imaging. Further optimization of this approach could support more efficient and accurate diagnostic workflows in clinical settings.

## Supporting information

**S1 Fig. Two cases diagnosed as chronic sinusitis with dystrophic calcifications (Class 2).** The algorithm misinterpreted these cases as fungal calcifications (Class 3). a) The lowest portion of a large calcification appears as punctate calcifications (red arrows). b) A small calcification was found adjacent to a large dense calcification (red arrows). (TIF)

## Author contributions

**Conceptualization:** Zepa Yang, Inseon Ryoo.

**Data curation:** Zepa Yang, Hoo Yun, Siwoo Kim, Hye Na Jung, Sangil Suh, Bo Kyu Kim, Byungjun Kim, Sung-Hye You, Inseon Ryoo.

**Formal analysis:** Zepa Yang, Hye Na Jung.

**Funding acquisition:** Zepa Yang, Inseon Ryoo.

**Investigation:** Insung Choi, Inseon Ryoo.

**Methodology:** Zepa Yang, Insung Choi, Hoo Yun, Siwoo Kim, Sung-Hye You, Inseon Ryoo.

**Project administration:** Inseon Ryoo.

**Resources:** Inseon Ryoo.

**Software:** Zepa Yang, Insung Choi.

**Supervision:** Zepa Yang, Inseon Ryoo.

**Validation:** Zepa Yang, Insung Choi.

**Visualization:** Zepa Yang.

**Writing – original draft:** Zepa Yang, Inseon Ryoo.

**Writing – review & editing:** Zepa Yang, Inseon Ryoo.

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
