## [Decision Letter · Decision Letter 0]

13 Oct 2025

Dear Dr. Ryoo,

Thank you for submitting your manuscript to PLOS ONE. After careful consideration, we feel that it has merit but does not fully meet PLOS ONE’s publication criteria as it currently stands. Therefore, we invite you to submit a revised version of the manuscript that addresses the points raised during the review process.

We look forward to receiving your revised manuscript.

Kind regards,

Rong-San Jiang

Academic Editor

PLOS ONE

**Journal Requirements:**

1. When submitting your revision, we need you to address these additional requirements. Please ensure that your manuscript meets PLOS ONE's style requirements, including those for file naming. The PLOS ONE style templates can be found at https://journals.plos.org/plosone/s/file?id=wjVg/PLOSOne_formatting_sample_main_body.pdf and https://journals.plos.org/plosone/s/file?id=ba62/PLOSOne_formatting_sample_title_authors_affiliations.pdf 2. Please note that PLOS One has specific guidelines on code sharing for submissions in which author-generated code underpins the findings in the manuscript. In these cases, we expect all author-generated code to be made available without restrictions upon publication of the work. Please review our guidelines at https://journals.plos.org/plosone/s/materials-and-software-sharing#loc-sharing-code and ensure that your code is shared in a way that follows best practice and facilitates reproducibility and reuse. 3. If the reviewer comments include a recommendation to cite specific previously published works, please review and evaluate these publications to determine whether they are relevant and should be cited. There is no requirement to cite these works unless the editor has indicated otherwise. 

**Additional Editor Comments:**

Please revise your manuscript based on the reviewers' comments.

Reviewers' comments:

**Comments to the Author**

1. Is the manuscript technically sound, and do the data support the conclusions?

Reviewer #1: Partly

Reviewer #2: Yes

2. Has the statistical analysis been performed appropriately and rigorously?

Reviewer #1: No

Reviewer #2: Yes

3. Have the authors made all data underlying the findings in their manuscript fully available?

Reviewer #1: No

Reviewer #2: Yes

4. Is the manuscript presented in an intelligible fashion and written in standard English?

Reviewer #1: Yes

Reviewer #2: Yes

**Reviewer #1:** 1. The definition of Class 1 is problematic. The segmentation and detection models are trained to find the sinus and then look for calcifications. However, the classification model's first class includes sinuses that, by definition, have no calcifications to detect. This creates a logical inconsistency. How does the classification CNN process a "bounding box" from YOLO if YOLO found nothing? It's crucial to clarify this workflow. Were "negative" patches also extracted and fed to the CNN for Class 1 training? If not, there is a high risk of label leakage, where the model learns to classify based on the mere presence of a YOLO proposal rather than the image features within it. This could explain the anomalously high recall and the low AUROC for Class 1.

2. Table 1 provides a good start but needs expansion. Most critically, it must include the number of cases/sinuses per class (1, 2, 3) for the training and each test set. The text mentions Class 2 data is limited, but quantifying this is essential for readers to assess potential class imbalance, which is a likely contributor to the lower performance on Class 2 . Adding details on the clinical diagnosis would also strengthen the dataset description.

3. The manuscript reports Weighted_accuracyalongside overall accuracy. However, the method for calculating this metric is not defined. Is it the balanced accuracy? Is it weighted by class support? Given the discussed class imbalance, clarifying this metric is vital for a correct interpretation of the results. The high weighted accuracy suggests good performance despite imbalance, but the reader needs to know how it was derived.

4.The discussion of false positives is good, but the analysis can be deeper. A supplementary table or figure showing examples of common misclassifications would be very valuable. For instance, what do Class 2 cases that are misclassified as Class 3 look like? This analysis is crucial for understanding the model's limitations and guiding future improvements. It directly addresses the weakness identified in the Kim et al. (2022) study that you correctly point out.

5. The study notes that YOLOv5's initial classification performance was poor, leading to the addition of a separate CNN for refinement. This warrants a brief discussion: Why was YOLOv5 chosen over other object detectors that might have stronger classification capabilities? Acknowledging that YOLO is optimized for speed and that its classification head can be weaker in complex medical imaging tasks would provide a more nuanced justification for the two-stage detection+classification design.

6. Expand the scope of the paper by referencing relevant works such as ” SkinDWNet: a novel deep learning model for multiclass classification of skin cancers using dermoscopic images”," CDC_Net: Multi-classification convolutional neural network model for detection of COVID-19, pneumothorax, pneumonia, lung Cancer, and tuberculosis using chest X-rays ", " SCDNet: a deep learning-based framework for the multiclassification of skin cancer using dermoscopy images ", and " DMFL_Net: A federated learning-based framework for the classification of COVID-19 from multiple chest diseases using X-rays, “Blockchain-federated and deep-learning-based ensembling of capsule network with incremental extreme learning machines for classification of COVID-19 using CT scans.

7. Figure 1 description in the text and the figure caption are too generic. The caption should specifically describe the proposed 3D U-Net architecture (e.g., number of layers, filter sizes, as described in the text) rather than a standard U-Net diagram.

In Table 2 & 3 abbreviation "DSC score = standard deviation" in Table 2 is incorrect . This must be corrected. In Table 3, consider adding a column for the number of samples per class in each test set to provide context for the performance metrics.

The low AUROC for Class 1 needs a more thorough discussion. Is it truly due to thresholding, or does it indicate that the model cannot reliably distinguish a "no calcification" patch from background noise or other non-calcification structures?

**Reviewer #2:** 1. The study data were all from the affiliated hospital of Korea University, and there may be selection bias, which will limit the universality and extrapolation of the results. It is recommended to clearly state this point in the discussion.

2. There are many types of fungal sinusitis (such as invasive, non-invasive, allergic fungal sinusitis, etc.). It is necessary to clearly state which type was collected in this study, otherwise the reader will not be able to judge the scope of application of the results.

3. There are many types of fungal sinusitis. The article shows that there are more male patients. Is this gender distribution applicable to all types of fungal sinusitis? The gender distribution of some types of fungal infections may vary in different regions or populations. The type of infection and its regional characteristics of the study subjects should be explained.

4. Were the images classified according to the results of pathological examination and bacterial culture? This can improve the accuracy of diagnosis and support the correlation between images and real pathology.

5. This article only includes cases of fungal infection with calcification. If there is fungal infection without calcification, is there a potential risk of missed diagnosis by this detection mode? The clinical significance and possible impact of this range limitation need to be discussed.

6. The study only considered CT calcification, and did not include other imaging features such as bone erosion, etc. However, calcification is usually easier to judge and not easy to miss. The advantage of this judgment is not obvious.

**Do you want your identity to be public for this peer review?** For information about this choice, including consent withdrawal, please see our Privacy Policy

Reviewer #1: **Yes:** Dr. Tayyaba Anees

Reviewer #2: No

---

## [Author Response · Author response to Decision Letter 1]

28 Oct 2025

Dear Editor and Reviewers

We sincerely thank you for your valuable time and insightful comments on our manuscript entitled “Deep Learning Detection and Classification of Fungal and Non-fungal Calcifications on Paranasal Sinus CT Imaging” (Manuscript ID PONE-D-25-33924).

We greatly appreciate the time and effort invested in reviewing our work. In response, we have carefully revised the paper in accordance with the reviewers’ suggestions. Below, we provide detailed responses to each comment. All changes are indicated in the revised manuscript (tracked version). We hope that the revisions meet the expectations of the reviewers and editors, and we kindly ask for your favorable consideration for the publication of our work.

Reviewer #1:

1-1. The definition of Class 1 is problematic. The segmentation and detection models are trained to find the sinus and then look for calcifications. However, the classification model's first class includes sinuses that, by definition, have no calcifications to detect. This creates a logical inconsistency. How does the classification CNN process a "bounding box" from YOLO if YOLO found nothing? It's crucial to clarify this workflow. Were "negative" patches also extracted and fed to the CNN for Class 1 training? If not, there is a high risk of label leakage, where the model learns to classify based on the mere presence of a YOLO proposal rather than the image features within it. This could explain the anomalously high recall and the low AUROC for Class 1.

 Thank you for this important observation.

In the revised manuscript, we have clarified the labeling procedure and data flow for Class 1 in the Detection and Classification Model section. All YOLO-detected bounding boxes and adjacent sinus regions were carefully reviewed by two board-certified radiologists. During this process, cases of chronic or non-fungal sinusitis without visible calcification were explicitly assigned as Class 1 (no calcification) under radiologists’ supervision. This ensured that the classifier was trained on representative non-calcified sinus patterns confirmed by experts.

Furthermore, the YOLO detector was trained with a relatively low confidence threshold to reduce false negatives, allowing subtle or ambiguous regions to be proposed for review and subsequent classification. This workflow prevents label leakage and follows cascaded frameworks commonly used in medical imaging studies for lesion detection and classification [1–4]. Corresponding revisions have been made to both the Methods and Discussion sections to clarify these points.

(M&M, L178-183, and discussion L 431-440).

1-2. Table 1 provides a good start but needs expansion. Most critically, it must include the number of cases/sinuses per class (1, 2, 3) for the training and each test set. The text mentions Class 2 data is limited, but quantifying this is essential for readers to assess potential class imbalance, which is a likely contributor to the lower performance on Class 2. Adding details on the clinical diagnosis would also strengthen the dataset description.

 We agree with you. We have expanded Table 1 to report the number of cases (n) and the number and proportion of sinuses (n, %) for Class 1, Class 2, and Class 3 for each cohort. To avoid ambiguity, we explicitly state that left and right sinuses are counted separately at the sinus level, while a case is defined as a unique study (ID + date). As a result, a single case can contribute different classes across sides; thus, the sum of class-specific case counts may exceed the total number of cases. These clarifications appear in the Table 1 legend/footnote and in Methods.

Because the reviewer specifically highlighted Class 2 (non-fungal calcification), the revised Table 1 now allows readers to quantify its relative scarcity across cohorts (Training, Internal, Temporal, and Geographic). We also added short notes in Discussion acknowledging the class imbalance—particularly the limited volume of Class 2—and its implications for interpreting performance metrics. (Discussion, L441-450)

1-3. The manuscript reports Weighted_accuracy alongside overall accuracy. However, the method for calculating this metric is not defined. Is it the balanced accuracy? Is it weighted by class support? Given the discussed class imbalance, clarifying this metric is vital for a correct interpretation of the results. The high weighted accuracy suggests good performance despite imbalance, but the reader needs to know how it was derived.

 We appreciate your comment and agree that the term “weighted accuracy” was ambiguous. In the revised version, we removed that label and explicitly described the accuracy measures we report in plain language:

- Overall accuracy (micro): the proportion of correctly classified sinuses out of all sinuses across all classes combined.

- Balanced accuracy (macro recall): the average of the per-class recalls—i.e., we compute the recall (sensitivity) separately for Class 1, Class 2, and Class 3 and then take the simple mean. This measure is less sensitive to class imbalance.

For transparency, the previously listed “weighted accuracy” corresponded to a support-weighted mean of per-class recall, which is numerically equivalent to overall accuracy in our setting and can therefore look optimistic when classes are imbalanced. To avoid confusion, we now report overall accuracy and balanced accuracy, together with per-class precision/recall/F1 and AUROC, and we clarify these definitions in the manuscript text and table footnotes. (M&M, L255-259 and Table 3)

1-4. The discussion of false positives is good, but the analysis can be deeper. A supplementary table or figure showing examples of common misclassifications would be very valuable. For instance, what do Class 2 cases that are misclassified as Class 3 look like? This analysis is crucial for understanding the model's limitations and guiding future improvements. It directly addresses the weakness identified in the Kim et al. (2022) study that you correctly point out.

 We appreciate your insightful comment. We have included supplementary images illustrating two cases of false positive class 2. Additionally, we have discussed about it in the Discussion. (Discussion, L416-419, suppl. Fig 1.)

1-5. The study notes that YOLOv5's initial classification performance was poor, leading to the addition of a separate CNN for refinement. This warrants a brief discussion: Why was YOLOv5 chosen over other object detectors that might have stronger classification capabilities? Acknowledging that YOLO is optimized for speed and that its classification head can be weaker in complex medical imaging tasks would provide a more nuanced justification for the two-stage detection+classification design.

 Thank you for the suggestion. We used YOLOv5 primarily as a high-recall proposal generator. In our data, calcifications are small, high-contrast targets; YOLOv5 offered reliable localization and fast training/inference with mature tooling and pretrained weights, which fit our CT slice pipeline and deployment needs. However, in preliminary tests its single-stage classification head was weaker for fine-grained discrimination between non-fungal vs. fungal calcifications—an observation consistent with YOLO’s emphasis on speed and objectness rather than subtle class separation in medical imaging.

To address this, we adopted a two-stage design: YOLOv5 is tuned for sensitivity to produce candidate regions, and a dedicated CNN performs fine-grained class refinement (Classes 1/2/3). This separation let us apply class-aware sampling/augmentation and loss choices to better handle imbalance, stabilize Class-2 performance, and keep latency practical. Alternative detectors could be used, but in our setting the high-recall proposals + specialized classifier trade-off was the most dependable. We have added this rationale to the Discussion with relevant references. (Discussion L376-384)

1-6. Expand the scope of the paper by referencing relevant works such as ” SkinDWNet: a novel deep learning model for multiclass classification of skin cancers using dermoscopic images”," CDC_Net: Multi-classification convolutional neural network model for detection of COVID-19, pneumothorax, pneumonia, lung Cancer, and tuberculosis using chest X-rays ", " SCDNet: a deep learning-based framework for the multiclassification of skin cancer using dermoscopy images ", and " DMFL_Net: A federated learning-based framework for the classification of COVID-19 from multiple chest diseases using X-rays, “Blockchain-federated and deep-learning-based ensembling of capsule network with incremental extreme learning machines for classification of COVID-19 using CT scans

 We appreciate the suggestions and reviewed the cited papers. After evaluation, we decided to add two references (CDC_Net and DMFL-Net as reference 11 and 12) as examples of previously published AI based research in radiology. However, most of the suggested references are misaligned with our problem setting—namely, lesion-level detection plus fine-grained classification of calcifications on paranasal sinus CT using a two-stage pipeline (YOLO proposals followed by CNN refinement). First, there is a modality and anatomy mismatch: SkinDWNet and SCDNet are dermoscopy studies (skin lesions), while CDC_Net and DMFL_Net are predominantly chest radiography. Our work concerns CT of the paranasal sinuses with calcification-focused targets and a detection-first workflow. Second, there is a task mismatch: the suggested works primarily address global image multi-class classification, whereas our study centers on lesion localization with low-threshold proposals and refined categorization at the sinus/lesion level. Third, there is a methodological scope mismatch: the blockchain-federated/capsule-network paper emphasizes federated learning, blockchain, and ensemble mechanics in COVID-19 CT classification, which is orthogonal to our objectives and risks confusing the methodological focus of a detection-then-refinement pipeline for sinonasal calcifications.

We have already included multiple multi-class medical-imaging examples where they are directly relevant, and we emphasize domain-specific prior work that is CT-based, calcification- or sinonasal-disease–related, and cascaded detection-to-classification in design. For these reasons, we could not add all the suggested dermoscopy, or blockchain-federated papers that would dilute the scope for readers of a sinonasal-CT lesion-level study.

1-7. Figure 1 description in the text and the figure caption are too generic. The caption should specifically describe the proposed 3D U-Net architecture (e.g., number of layers, filter sizes, as described in the text) rather than a standard U-Net diagram.

In Table 2 & 3 abbreviation "DSC score = standard deviation" in Table 2 is incorrect . This must be corrected. In Table 3, consider adding a column for the number of samples per class in each test set to provide context for the performance metrics.

The low AUROC for Class 1 needs a more thorough discussion. Is it truly due to thresholding, or does it indicate that the model cannot reliably distinguish a "no calcification" patch from background noise or other non-calcification structures?

 Thank you for pointing these out. We revised the caption to describe the proposed 3D U-Net specifically, including the number of down-sampling levels (N = 4), kernel sizes (3×3×3 and 1×1×1), and encoder channel widths (32, 64, 128, 256). We also updated the first in-text reference to Fig 1 in Methods to reflect these architectural details. (Legend of Fig. 1)

We corrected the abbreviation in Table 2: “DSC” now stands for Dice Similarity Coefficient (SD denotes standard deviation throughout). In Table 3, we added a column reporting the number of samples per class in each test cohort at the sinus level (left/right counted separately), so that readers can contextualize the performance metrics.

We expanded the Discussion to give a clearer interpretation. Class 1 (no abnormal calcification) showed a relatively low AUROC (0.54). This reflects our low detection threshold in the proposal stage, which increases the number of non-calcification candidates passed to the classifier and can raise the false-positive rate for the negative class at some operating points. Negative patches are also heterogeneous (for example, mucosal thickening, small bony structures, dental or metallic artifacts), which makes the decision boundary broader than for calcification-positive classes. We added a note that class-specific operating points and probability calibration may improve Class-1 behavior. (Discussion, L431-440)

Reviewer #2:

2-1. The study data were all from the affiliated hospital of Korea University, and there may be selection bias, which will limit the universality and extrapolation of the results. It is recommended to clearly state this point in the discussion.

 Thank you for your comment. Although the images in the external test set were collected from a different branch of the same university-affiliated hospital system, the practicing physicians and radiologists, as well as the CT machines and imaging protocols, were different. Consequently, we had to adjust all relevant parameters in order to apply the trained algorithm to that dataset. Moreover, since the two hospitals are located in opposite parts of the city (northeast vs. southwest), the patient populations they serve differ significantly. We have now added these details to the “Data Acquisition” subsection of the M & M.(M&M, L92-97)

2-2. There are many types of fungal sinusitis (such as invasive, non-invasive, allergic fungal sinusitis, etc.). It is necessary to clearly state which type was collected in this study, otherwise the reader will not be able to judge the scope of application of the results.

 We appreciate your valuable comment. Our study focused on non-invasive fungal sinusitis, specifically fungal ball, which typically presents with characteristic calcification patterns on radiographic images. We corrected terms or added “fungal ball” all through the manuscript.

Allergic fungal sinusitis and invasive fungal sinusitis (both acute and chronic forms) were not included in this study, and we have noted this limitation in the Discussion section. (Discussion, L485-494)

Allergic fungal sinusitis often demonstrates hyperintense sinus contents, which can overlap with findings of other inflammatory sinus diseases containing proteinous or mucinous material. In contrast, invasive fungal sinusitis is characterized by bony destruction and direct invasion into adjacent structures—features that were beyond the scope of our current evaluation. We agree that future studies incorporating these types of fungal sinusitis will be valuable to provide a more comprehensive understanding of fungal sinus diseases.

2-3. There are many types of fungal sinusitis. The article shows that there are more male patients. Is this gender distribution applicable to all types of fungal sinusitis? The gender distribution of some types of fungal infections may vary in different regions or populations. The type of infection and its regional characteristics of the study subjects should be explained.

 Thank you for your insightful comment. You are correct that the gender distribution varies depending on the type of fungal sinusitis. Previous studies have reported a female predominance in cases of fungal ball and a male predominance in invasive fungal sinusitis. Since our study focused on fungal ball cases, a female predominance was observed across all datasets. We identified that “female” and “male” were mistakenly written in reverse in the table, and we have corrected this error accordingly. We appreciate your careful observation. (table 1)

2-4. Were the images classified according to the results of pathological examination and bacterial culture? This can improve the accuracy of diagnosis and support the correlation between images and real pathology.

 Thank you for your keen suggestion. The diagnosis of fungal calcifications was confirmed by pathological examination of specimens obtained from endoscopic sinus surgery. Pathologists diagnosed fungal ball with specific findings such as fungal hyphae. This info

---

## [Decision Letter · Decision Letter 1]

28 Dec 2025

Deep Learning Detection and Classification of Fungal and Non-fungal Calcifications on Paranasal Sinus CT Imaging

PONE-D-25-33924R1

Dear Dr. Ryoo,

We’re pleased to inform you that your manuscript has been judged scientifically suitable for publication and will be formally accepted for publication once it meets all outstanding technical requirements.

Kind regards,

Mohmed Isaqali Karobari, BDS, MScD.Endo, Ph.D. Endo, FDS, FPFA, FICD, MFDS

Academic Editor

PLOS One

Additional Editor Comments (optional):

Dear Authors,

The authors have addressed all the comments and suggestions provided by the reviewers, and the manuscript has undergone significant improvement. I would like to congratulate the authors and wish them all the very best in their future endeavours.

Best regards and keep well.

Reviewers' comments:

Reviewer's Responses to Questions

**Comments to the Author**

Reviewer #1: All comments have been addressed

Reviewer #2: All comments have been addressed

2. Is the manuscript technically sound, and do the data support the conclusions?

Reviewer #1: Yes

Reviewer #2: Partly

3. Has the statistical analysis been performed appropriately and rigorously?

Reviewer #1: Yes

Reviewer #2: Yes

4. Have the authors made all data underlying the findings in their manuscript fully available?

Reviewer #1: Yes

Reviewer #2: Yes

5. Is the manuscript presented in an intelligible fashion and written in standard English?

Reviewer #1: Yes

Reviewer #2: Yes

Reviewer #1: As per my evaluation Suggested changes have been addressed by the authors in the paper so it can be accepted.

Reviewer #2: Dear Editor, I believe the authors have fully responded to the reviewers' concerns, and the article is acceptable.

**Do you want your identity to be public for this peer review?** For information about this choice, including consent withdrawal, please see our Privacy Policy

Reviewer #1: **Yes:** Dr. Tayyaba Anees

Reviewer #2: No

---

## [Editor Report · Acceptance letter]

PONE-D-25-33924R1

PLOS One

Dear Dr. Ryoo,

I'm pleased to inform you that your manuscript has been deemed suitable for publication in PLOS One. Congratulations! Your manuscript is now being handed over to our production team.

Kind regards,

on behalf of

Prof Dr. Mohmed Isaqali Karobari

Academic Editor

PLOS One